# A high-throughput yeast display approach to profile pathogen proteomes for MHC-II binding

**Brooke D Huisman[1,2], Zheng Dai[3,4], David K Gifford[2,3,4], Michael E Birnbaum[1,2,5]***

[1]Koch Institute for Integrative Cancer Research, Cambridge, United States; [2]Department of Biological Engineering, Massachusetts Institute of Technology, Cambridge, United States; [3]Computer Science and Artificial Intelligence Laboratory, Massachusetts Institute of Technology, Cambridge, United States; [4]Department of Electrical Engineering and Computer Science, Massachusetts Institute of Technology, Cambridge, United States; [5]Ragon Institute of MGH, MIT and Harvard, Cambridge, United States

**Abstract** T cells play a critical role in the adaptive immune response, recognizing peptide antigens presented on the cell surface by major histocompatibility complex (MHC) proteins. While assessing peptides for MHC binding is an important component of probing these interactions, traditional assays for testing peptides of interest for MHC binding are limited in throughput. Here, we present a yeast display-based platform for assessing the binding of tens of thousands of user-defined peptides in a high-throughput manner. We apply this approach to assess a tiled library covering the SARS-CoV-2 proteome and four dengue virus serotypes for binding to human class II MHCs, including HLA-DR401, -DR402, and -DR404. While the peptide datasets show broad agreement with previously described MHC-binding motifs, they additionally reveal experimentally validated computational false positives and false negatives. We therefore present this approach as able to complement current experimental datasets and computational predictions. Further, our yeast display approach underlines design considerations for epitope identification experiments and serves as a framework for examining relationships between viral conservation and MHC binding, which can be used to identify potentially high-interest peptide binders from viral proteins. These results demonstrate the utility of our approach to determine peptide-MHC binding interactions in a manner that can supplement and potentially enhance current algorithm-based approaches.

*For correspondence:
mbirnb@mit.edu

## Editor's evaluation

This paper will be of interest to immunologists who study T cell recognition and informaticians who predict peptide ligands for MHC. It rigorously innovates a high-throughput approach to identify MHC ligands of pathogens on the proteome level using a previously developed yeast display-based platform, including ligands not identified by computational prediction. The data support the conclusions well and open a direction for future work to address the immunological significance of the findings.

## Introduction

Major histocompatibility complex (MHC) proteins play a critical role in adaptive immunity by presenting peptide fragments on the surface of cells. Peptide-MHCs (pMHCs) are then surveilled by T cells via their T cell receptors (TCRs), enabling immune cells to sense dysfunction, such as the

presence of pathogen-derived peptides (*Chaplin, 2010*; *Hennecke and Wiley, 2001*). Class II MHC molecules (MHC-II) are expressed primarily on professional antigen-presenting cells, and are recognized by antigen-specific CD4$^+$ T cells that drive the coordination of innate and adaptive immune responses (*Chaplin, 2010*; *Swain et al., 2012*). MHC-II molecules have an open peptide-binding groove, allowing for display of long peptides, consisting of a nine amino acid 'core' flanked by a variable number of additional residues on each side (*Jones et al., 2006*).

Generating reliable and rapid data on peptide-MHC binding is beneficial for understanding the underlying biology of adaptive immunity and for clinical applications, including for optimized T cell epitopes in vaccine design (*Dai et al., 2021*; *Keskin et al., 2019*; *Liu et al., 2020*; *Liu et al., 2021b*; *Moise et al., 2015*; *Ott et al., 2017*; *Patronov and Doytchinova, 2013*; *Rosati et al., 2021*). In fact, therapeutics to generate antigen-specific T cell responses have shown great promise in cancer (*Keskin et al., 2019*; *Ott et al., 2017*) and infectious disease (*Gambino et al., 2021*). Since understanding peptide-MHC binding is critical for identifying and engineering T cell epitopes, there have been sustained efforts to produce high-quality experimental data and predictive algorithms.

Initial experimental methods for determining peptide binding to MHC relied upon the analysis of synthesized candidate peptides via MHC stability or functional assays, and can produce high-confidence data, but can be difficult to scale beyond a small number of candidate peptides (*Altmann and Boyton, 2020*; *Justesen et al., 2009*; *Mateus et al., 2020*; *Sidney et al., 2010*; *Yin and Stern, 2014*). More recently, mass spectrometry-based approaches have been demonstrated for determining the MHC-presented peptide repertoire of cells. These approaches include monoallelic mass spectrometry, which allows for the unambiguous assignment of presented peptides to a given MHC allele. However, mass spectrometry-based approaches are not necessarily quantitative measures of presented peptide affinity or abundance, although there have been advances in quantitation using internal standards (*Stopfer et al., 2021*; *Stopfer et al., 2020*). Additionally, the peptides endogenously expressed by a cell can crowd out exogenously examined peptides of interest, and mass spectrometry approaches typically require large numbers of input cells (*Abelin et al., 2019*; *Abelin et al., 2017*; *Parker et al., 2021*; *Purcell et al., 2019*).

A wave of higher throughput approaches have been recently developed for studying peptide-MHC interactions, including yeast display (*Jiang and Boder, 2010*; *Liu et al., 2021a*; *Rappazzo et al., 2020*; *Wen et al., 2008*) and mammalian display-based methods (*Obermair et al., 2022*). Many of these assays rely upon construction of DNA-based libraries (*Jiang and Boder, 2010*; *Obermair et al., 2022*; *Rappazzo et al., 2020*; *Wen et al., 2008*), although approaches using chemically synthesized peptides have also recently been described (*Liu et al., 2021a*; *Smith et al., 2019*). DNA libraries have been generated either via DNA oligonucleotide synthesis (*Jiang and Boder, 2010*; *Obermair et al., 2022*; *Rappazzo et al., 2020*) or random fragmentation and insertion of viral genomic material (*Wen et al., 2008*). Upon assembly of the peptide libraries, peptide stabilization and surface expression (*Jiang and Boder, 2010*; *Obermair et al., 2022*; *Wen et al., 2008*) or peptide dissociation (*Rappazzo et al., 2020*) were used to assess peptide-MHC binding.

In addition to experimental advances, computational approaches for peptide-MHC binding prediction have advanced markedly over the past decade. These developments are due to algorithmic advances (*O'Donnell et al., 2020*; *Racle et al., 2019*; *Reynisson et al., 2020*; *Zeng and Gifford, 2019*) and the availability of large, high-quality training data (*Abelin et al., 2019*; *Abelin et al., 2017*; *Rappazzo et al., 2020*; *Reynisson et al., 2020*). However, despite the improvements in predicting peptide binding to MHC in a broad sense, the predictive power for individual peptides often remains imperfect relative to experimental measurements (*Rappazzo et al., 2020*; *Zhao and Sher, 2018*).

Here, we present a yeast display approach to directly assess peptide-MHC binding for large collections of defined peptide antigens to screen whole viral proteomes for MHC-II binding in high throughput. We utilize this approach to screen the full proteome of SARS-CoV-2, a present, global threat to public health, and identify and experimentally validate SARS-CoV-2-derived MHC binders, including both algorithmically predicted and algorithmically missed peptide binders, highlighting the potential of this approach to supplement or augment prediction algorithms. We additionally apply this approach to screen proteomes from serotypes 1–4 of dengue viruses, in which antibody-dependent enhancement results in more severe disease upon second infection with a different dengue virus serotype (*Guzman et al., 2016*), and thus represents a potential important application area for T cell-directed therapeutics. Our approach enables exploration of peptide binding to

MHCs in the context of serotype-specific mutations, identifying homologous, pan-serotype regions of interest that are capable of MHC binding and thus may represent desirable targets for immune interventions.

## Results

### Generation of yeast display libraries for profiling the SARS-CoV-2 proteome

Previous studies have reported the use of yeast-displayed MHC-II for characterizing peptide-MHC and pMHC-TCR interactions (*Rappazzo et al., 2020*; *Birnbaum et al., 2014*; *Rappazzo et al., 2020*, *Fernandes et al., 2020*). We adapted MHC-II yeast display constructs (*Rappazzo et al., 2020*) to generate a defined library of peptides that cover the SARS-CoV-2 proteome to assess them for MHC binding. To compare SARS-CoV-2 with a related coronavirus, we also included peptides from the spike and nucleocapsid proteins from SARS-CoV.

Each protein was windowed into peptides of 15 amino acids in length, with a step size of 1 to cover every possible 15mer peptide in the protein (*Figure 1a*). Each peptide was encoded in DNA and cloned in a pooled format into yeast vectors containing MHC-II proteins. The generated library was linked to three MHC-II alleles: HLA-DR401 (HLA-DRA1*01:01, HLA-DRB1*04:01), HLA-DR402 (HLA-DRA1*01:01, HLA-DRB1*04:02), and HLA-DR404 (HLA-DRA1*01:01, HLA-DRB1*04:04). Yeast were formatted with a flexible linker connecting the peptide and MHC, containing a 3C protease site and a Myc epitope tag, which can be used for selections (*Figure 1a*; *Rappazzo et al., 2020*). The final library contained 11,040 unique peptides, with 99% of the designed peptides present in each cloned yeast library, as assessed by next-generation sequencing.

### Strategies for selecting defined libraries

To enrich for peptide binders, iterative selections were performed (*Figure 1a*): the library is first incubated with competitor peptide and 3C protease, which cleaves the covalent linkage between peptide and MHC, followed by the addition of HLA-DM at lower pH. These conditions allow for the encoded peptide to be displaced from the peptide-binding groove. The Myc epitope tag is proximal to the peptide, which can be identified via incubation with an anti-epitope tag antibody followed by enrichment via magnetic bead selection if the yeast-expressed peptide remains bound to the MHC after the peptide exchange reaction.

Three rounds of selection were iteratively performed. Representative enrichment of yeast expressing Myc-tagged peptides can be seen in *Figure 1c* ('undoped library'), for the library displayed by HLA-DR401. Here, the pre-selection Myc-positive population starts at 29.3% and quickly converges, with 65.0% positive in the pre-selection Round 2 population and 74.1% in the pre-selection Round 3 population.

Given the rapid convergence of the library, we performed a second set of selections in which we doped the defined library into a randomized, null library to enable a greater degree of enrichment as compared to non-binding peptides. The null library was generated by fully randomizing 10 amino acids in the peptide region of the peptide-MHC-II construct while fixing three amino acids to encode stop codons. This library provides a baseline population of yeast which should not express pMHC, and therefore not enrich in our selections. We doped our defined peptide library into a 500-fold excess of null library, such that each peptide member was represented at approximately the same frequency (*Figure 1b*). The null library provides baseline competition, which true binders must enrich beyond, and increases the stringency of the enrichment task.

We performed four rounds of selection on the doped library. Because of the excess of null yeast, the initial pre-selection stain is low (1.6%) compared to the initial undoped library (*Figure 1c*). This staining enriched over the first three rounds of selection, reflective of the stringency of the task and clarity of enrichment. This is in contrast to the initial undoped library, which began with a much higher pre-selection stain, with a lower fold change in staining over rounds of selection. The low frequency of each member in the starting doped library, however, increases the likelihood of stochastic dropout for any given member.

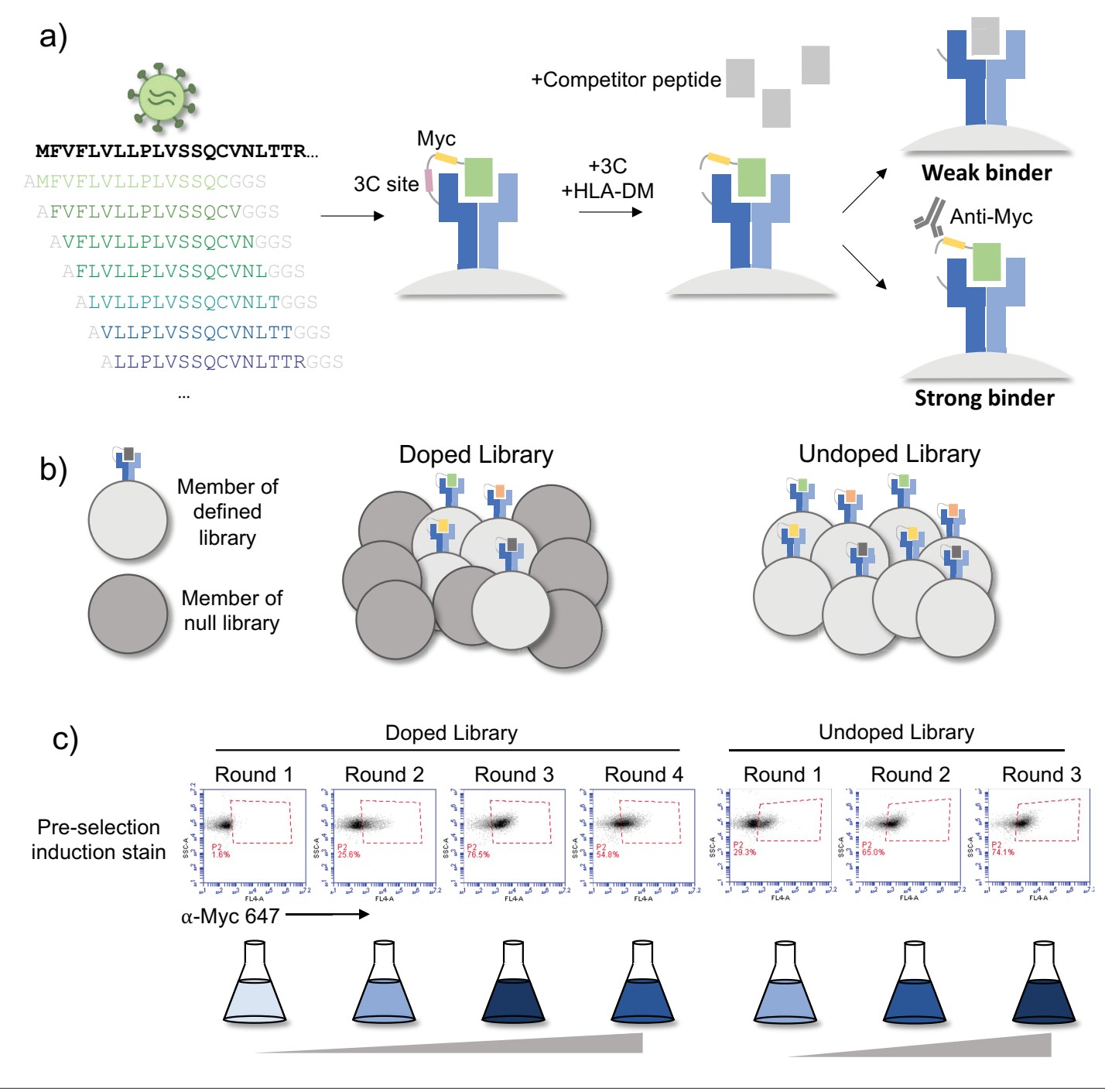

**Figure 1.** Overview of library and selections. (**a**) The defined library contains pathogen proteome peptides (length 15, sliding window 1). Poor binding peptides are displaced with addition of protease, competitor peptide, and HLA-DM. (**b**) Schematic of doped and undoped libraries: in the doped selection strategy, the library is added to a library of null, non-expressing constructs. (**c**) Representative flow plots showing enrichment of MHC-expressing yeast over rounds of selection for the library containing SARS-CoV-2 and SARS-CoV peptides on HLA-DR401.

The online version of this article includes the following figure supplement(s) for figure 1:

**Figure supplement 1.** Correlations between selection rounds.

## Analysis of selection data

After selections, peptide identities were determined through deep sequencing of enriched yeast populations, providing us with a dataset comprised of positive enrichment over four rounds of selection from the doped library and both positive and negative enrichment for three rounds of selection from the undoped library (*Supplementary file 1a*). *Figure 1—figure supplement 1* shows the correlation between defined library members on HLA-DR401. As expected, the unselected library correlated poorly with post-selection rounds. Consistent with the observed staining (*Figure 1c*), the doped library essentially converged after Round 3. Similarly, the undoped library appears converged following Round 2.

Next, we established metrics for enrichment for each mode of selection. Given the high starting frequency of members in the undoped library, we classify enrichment based on fold change between Round 1 and Round 2, and we define criteria for enriched yeast in the undoped library as making up a higher fraction of reads following Round 2 compared to Round 1. In contrast, in the doped library, members start at low frequencies, and we define enrichment based on presence above a threshold in Round 3 of selection, specifically as having greater than or equal to 10 reads following Round 3. *Figure 2b* illustrates the correspondence between enrichment metrics in the doped and undoped library for the library on HLA-DR401. Of the 11,040 peptides in the library, 2467 enriched in both the doped and undoped libraries displayed by HLA-DR401 (*Figure 2a*). An additional 1252 enriched in the doped library only and 797 enriched in the undoped library only.

Because the library is designed with a step size of 1, we next utilized overlap between adjacent peptides to determine high-confidence binders. This analysis allows us to address the potential that peptide sequences could register shift in such a way that invariant portions of the linker sequences could inadvertently be incorporated into the peptide-binding groove. To do this, we develop and implement a smoothing method, examining overlapping peptides for shared enrichment behavior. Classically, the strongest determinant of peptide affinity for an MHC is the nine amino acid stretch sitting within the peptide-binding groove (*Jones et al., 2006*; *Stern et al., 1994*), although proximal peptide flanking residues can also affect binding (*Lovitch et al., 2006*; *O'Brien et al., 2008*; *Zavala-Ruiz et al., 2004*). In our libraries, a given 9mer is present in seven overlapping 15mer peptides, and we calculate how many of these seven 15mers have enriched. This calculation is shown schematically in *Figure 2—figure supplement 1a* with toy sequences and applied to enrichment data for SARS-CoV-2 nucleocapsid on HLA-DR401 in *Figure 2—figure supplement 1b*. Sequences with good 9mer cores should enrich along with neighboring sequences with the same 9mer sequence. In contrast, sequences which enrich spuriously or due to linker sequence in the peptide groove or other stochastic factors should have few neighbor sequences also enriching. Thus, we define a cutoff for high-confidence 9mer enrichment of five out of seven 9mer-containing sequences enriching. This cutoff tolerates some stochastic dropout, while still disallowing any cores that may solely enrich by register shifting the Gly-Ser linker residues into the Position 9 pocket, which are favorable for each MHC allele in our study (*Abelin et al., 2019*; *Rappazzo et al., 2020*; *Reynisson et al., 2020*). Of the 2467 peptides which enriched in both the doped and undoped libraries for HLA-DR401, 1791 also contain a 9mer sequence which enriched in five or more peptides of the seven neighboring sequences containing it (*Figure 2a*), with 676 peptides enriching in both doped and undoped libraries but not containing a 9mer core enriched in five or more peptides, and 788 15mers containing a 9mer which enriched in five or more peptides but enriched in zero or one of the doped and undoped libraries. These full relationships are captured in Venn diagrams in *Figure 2—figure supplement 2* for all three MHC alleles studied here.

## Sequence motifs of enriched peptides are consistent with known binders and highlight considerations for designing epitope identification experiments

To examine the 9mer core motifs of enriched peptides, we utilized a position weight matrix (PWM) method to infer the peptide register and generated visualizations of the 9mer cores using Seq2Logo (*Thomsen and Nielsen, 2012*). *Figure 2c* shows a sequence logo of the aligned 9mer cores from the 2467 15mer peptides which enriched on HLA-DR401 in both doped and undoped libraries. The peptide motif is consistent with previously reported motifs for HLA-DR401 (*Abelin et al., 2019*; *Rappazzo et al., 2020*): hydrophobic amino acids are preferred at P1, acidic residues at P4, polar

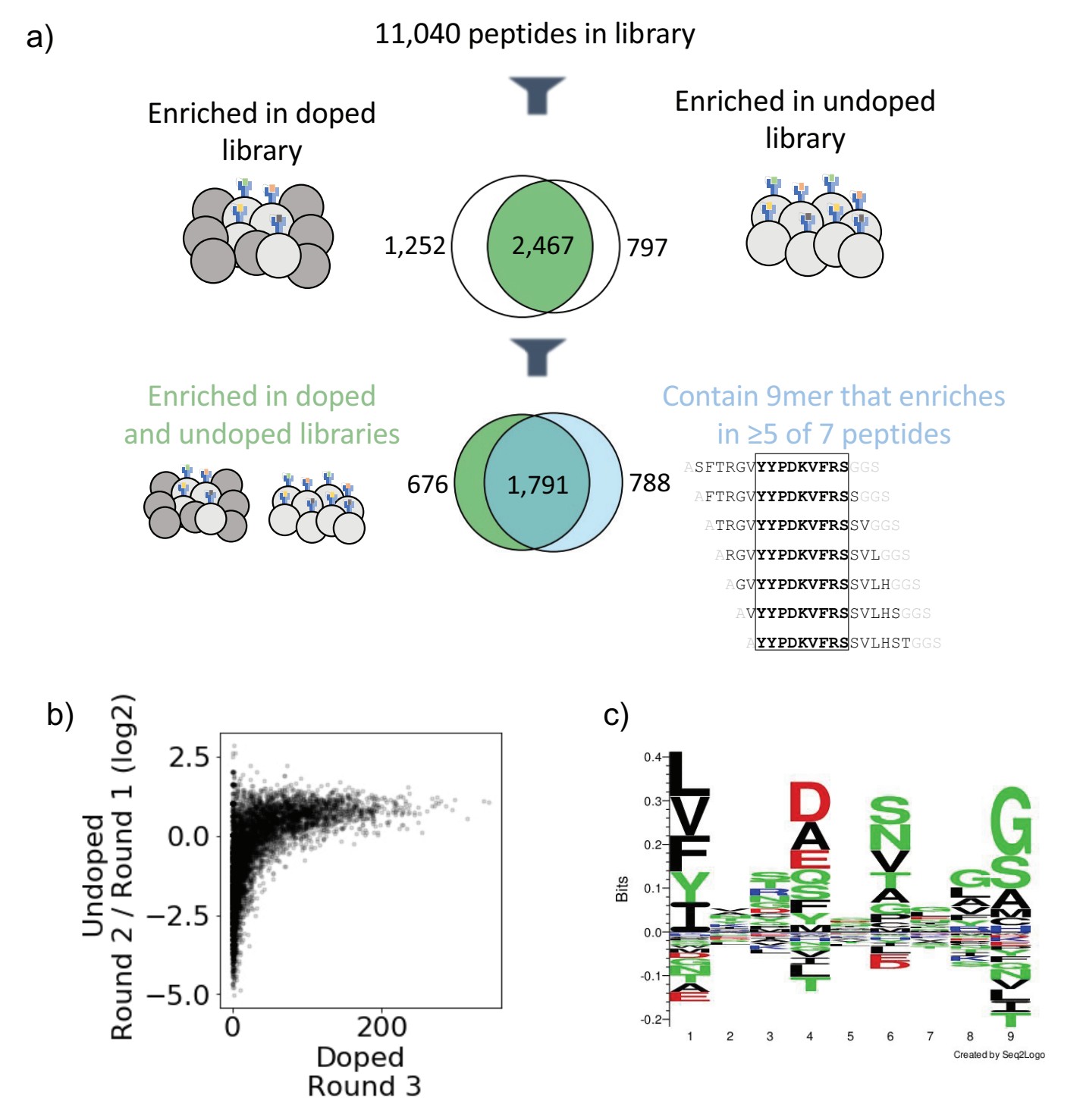

**Figure 2.** Output of selections and analysis of selection data. (**a**) Overview of filtering peptides and correspondence between selection strategies for SARS-CoV and SARS-CoV-2 library on HLA-DR401. Peptides are filtered for enrichment in both doped and undoped libraries. Further, the relationship between these peptides and peptides which contain a 9mer that is enriched in five or more of the seven peptides containing it is shown. (**b**) Relationships between enrichment in doped and undoped libraries. Absolute counts following Round 3 of selection of the doped library are plotted against the log2 fold change between read fraction for peptides in Round 2 and Round 1. Data are shown for the library on HLA-DR401. (**c**) Sequence logo of 2467 peptides that enriched in both doped and undoped selected libraries for HLA-DR401. Registers are inferred with a position weight matrix-based alignment method. Logos were generated with Seq2Logo-2.0.

The online version of this article includes the following figure supplement(s) for figure 2:

*Figure 2 continued on next page*

*Figure 2 continued*

**Figure supplement 1.** Utilizing overlapping peptides to call high-confidence peptides.

**Figure supplement 2.** Full Venn diagrams.

**Figure supplement 3.** Sequence logo for HLA-DR402 and HLA-DR404.

residues at P6, and small residues at P9. We also observe some preference for glycine at P8 in the sequence logo, which is potentially an artifact of non-native registers with linker at P8 and P9.

The other alleles used in the study, HLA-DR402 and HLA-DR404, have polymorphisms in their peptide-binding groove sequences as compared to HLA-DR401, which affect binding preferences. HLA-DR401 differs from HLA-DR402 at four amino acids and from HLA-DR404 at two amino acids, with

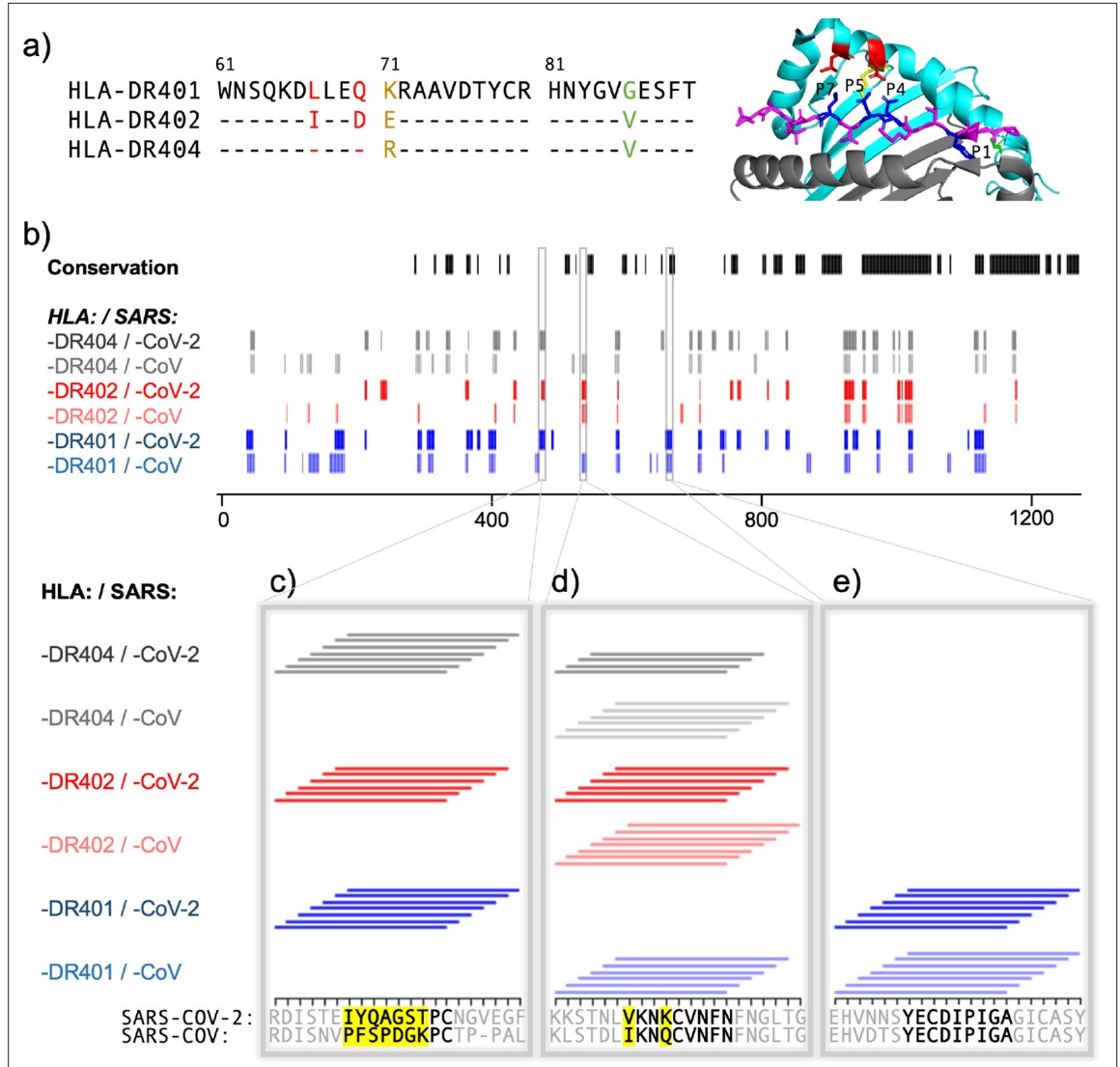

**Figure 3.** Comparing HLA-DR401, HLA-DR402, and HLA-DR404 for binding to related spike proteins from SARS-CoV-2 and SARS-CoV. (**a**) Sequence alignment showing sequence differences in HLA-DR402 and HLA-DR404 compared to HLA-DR401 and highlighted on HLA-DR401 structure (PDB 1J8H). Colors are: red for amino acids shared between HLA-DR401 and HLA-DR404, green for amino acids shared between HLA-DR402 and HLA-DR404, and yellow for amino acids different in all three alleles. Affected peptide positions (**P1, P4, P5, P7**) are colored in blue and labelled on the structure. (**b**) Conservation and enrichment of 9mer peptides from SARS-CoV-2 and SARS-CoV spike proteins. Conserved 9mers are indicated in black. If a 9mer along the proteome enriched in five or more of the adjacent peptides containing it, its enrichment is indicated with a vertical line with color for allele (HLA-DR401: blue; HLA-DR402: red; HLA-DR404: gray) and opacity for virus (SARS-CoV-2: dark; SARS-CoV: light). (**b–e**) Zoomed regions show enrichment of individual 15mer peptides. Only peptides containing the bolded 9mer sequence are shown. Amino acids in the bolded 9mer that are not conserved between SARS-CoV-2 and SARS-CoV are highlighted in yellow.

all polymorphisms located in the beta chain. HLA-DR402 and HLA-DR404 share an amino acid distinct from HLA-DR401 affecting the P1 pocket (Gly86Val), resulting in a preference for smaller hydrophobic residues (*Figure 3a*). Three polymorphisms in HLA-DR402 affect P4, P5, and P7 compared to HLA-DR401 (Leu67Ile, Gln70Asp, and Lys71Glu), while HLA-DR404 has only one (Lys71Arg). Sequence logos for HLA-DR402 and HLA-DR404 are consistent with previously reported motifs and MHC polymorphisms (*Figure 2—figure supplement 3*). For HLA-DR402, we observe less P4 preference compared to the motif of HLA-DR402 binders enriched from a randomized yeast display peptide library (*Rappazzo et al., 2020*), albeit consistent with mass spectrometry-generated motifs which also showed minimal P4 preference for HLA-DR402 (*Abelin et al., 2019*).

To explore differences between mass spectrometry, defined libraries, and random libraries, and to probe the differing strengths of P4 peptide preference observed for HLA-DR402 between these modalities, we examined the compositions of randomized and defined libraries. We hypothesized that skewed amino acid abundances in nature, which are reflected in the defined library, could result in an apparent diminished amino acid preference. Indeed, three of the most preferred P4 residues for binding HLA-DR402, Trp, His, and Met (*Rappazzo et al., 2020*), are all low abundance in the SARS-CoV-2 proteome (Trp 1.1%, His 1.9%, Met 2.2%). In comparison, a randomized peptide library for HLA-DR402 (*Rappazzo et al., 2020*) had a higher representation of these amino acids (Trp 3.8%, His 2.9%, Met 3.8%). Additionally, the randomized library had approximately 9000-fold more members than the defined library, providing more instances of all amino acids. The low abundance and under-representation of these amino acids likely underlies the apparent lack of amino acid consensus at P4 in enriched peptides. Interestingly, Arg and Lys, which have also been reported as preferred HLA-DR402 P4 residues, are more abundant than Trp, His, and Met in the SARS-CoV-2 proteome (Arg 3.4% and Lys 5.9%; compare to Arg 9.7%, Lys 4.0% in the random library), but still show less representation at P4 in the defined library enriched peptides compared to the random library-enriched peptides. These differences in motifs between randomized and defined libraries highlight the utility of randomized libraries for downstream applications such as training prediction algorithms. Approaches influenced by amino acid abundance in nature, such as defined libraries and mass spectrometry approaches, could inadvertently bias against possible binders because of absence of amino acids in their null distribution, rather than true binding preference.

Next, we wanted to examine the distribution of peptides among the possible 9mer registers along each 15 amino acid sequence. Based on our register inference, of the 2467 enriched peptides from the HLA-DR401 library, 1610 peptides bound native 9mer cores without using any linker sequence residues in the 9mer core, which is consistent with theoretical ratios of possible native and non-native cores for a given 9mer (*Supplementary file 1b*). The peptides with predicted native 9mer cores were approximately equally distributed between possible registers, with the exception of the N-terminal register, which had one-third fewer peptides. This register had only a single N-terminal flanking residue (a fixed Ala), which is likely disfavored.

Because the library was designed with step size of one, many of the 9mer cores will be repeated among neighboring peptides. Of the 1610 HLA-DR401 peptides which enriched using a native 9mer core, there are 563 unique 9mer cores identified through register inference. *Table 1* summarizes enrichment for each protein included in the library, highlighting the number of 15mers which enriched in both the doped and undoped libraries, the number of unique native 9mer cores, and the number of 15mers containing a 9mer enriched in at least five of seven overlapping peptides.

## Examining relationships between MHC-specific binding and spike proteins from SARS-CoV-2 and SARS-CoV

To further explore relationships between the MHCs studied here and their virally derived peptide repertoires, we compared the binding of SARS-CoV-2 and SARS-CoV spike proteins to all three MHC alleles. Sequence alignment of these three MHC alleles is shown in *Figure 3a*, with polymorphic regions highlighted on an HLA-DR401 structure (adapted from PDB 1J8H). Interplay between viral conservation and binding are illustrated in *Figure 3b*, highlighting conserved regions of the proteome in black and binders to each allele in gray, red, and blue. Regions are highlighted where sequences enrich in overlapping peptides; that is, for each nine amino acid stretch along the proteome, we calculated how many of the seven 15mer peptides enrich in the yeast display assay, and if a 9mer enriched five or more times, it is marked as a hit. Specific examples of these relationships are probed

**Table 1.** Summary of enriched peptides for each source protein, including: the number of unique 15mers which each enriched in both of the doped and undoped libraries; the number of unique 9mer cores identified by register inference in these enriched 15mers (native cores only, so linker-containing inferred cores excluded); and the number of unique enriched 15mers that contain 9mer sequences enriched in five or more of overlapping neighbors.

| Virus | Protein | Protein length (# of amino acids) | MHC allele | # of 15mers | # of 9mer cores | # of smoothed 15mers |
|---|---|---|---|---|---|---|
| SARS-CoV | Spike | 1255 | HLA-DR401 | 324 | 74 | 221 |
| | | | HLA-DR402 | 217 | 65 | 110 |
| | | | HLA-DR404 | 289 | 61 | 193 |
| SARS-CoV | Nucleocapsid | 422 | HLA-DR401 | 40 | 8 | 34 |
| | | | HLA-DR402 | 34 | 13 | 12 |
| | | | HLA-DR404 | 31 | 6 | 20 |
| SARS-CoV-2 | Spike | 1273 | HLA-DR401 | 305 | 67 | 221 |
| | | | HLA-DR402 | 230 | 62 | 130 |
| | | | HLA-DR404 | 290 | 64 | 217 |
| SARS-CoV-2 | Nucleocapsid | 419 | HLA-DR401 | 34 | 8 | 24 |
| | | | HLA-DR402 | 33 | 10 | 15 |
| | | | HLA-DR404 | 30 | 8 | 18 |
| SARS-CoV-2 | Replicase polyprotein 1ab | 7096 | HLA-DR401 | 1652 | 388 | 1204 |
| | | | HLA-DR402 | 1104 | 325 | 678 |
| | | | HLA-DR404 | 1368 | 350 | 890 |
| SARS-CoV-2 | Non-structural protein 8 | 121 | HLA-DR401 | 41 | 10 | 32 |
| | | | HLA-DR402 | 21 | 7 | 17 |
| | | | HLA-DR404 | 32 | 8 | 19 |
| SARS-CoV-2 | Protein 7a | 121 | HLA-DR401 | 27 | 8 | 18 |
| | | | HLA-DR402 | 7 | 3 | 0 |
| | | | HLA-DR404 | 13 | 2 | 6 |
| SARS-CoV-2 | Non-structural protein 6 | 61 | HLA-DR401 | 0 | 0 | 0 |
| | | | HLA-DR402 | 1 | 1 | 0 |
| | | | HLA-DR404 | 0 | 0 | 0 |
| SARS-CoV-2 | Membrane protein | 222 | HLA-DR401 | 40 | 7 | 29 |
| | | | HLA-DR402 | 26 | 6 | 19 |
| | | | HLA-DR404 | 23 | 7 | 21 |
| SARS-CoV-2 | Envelope small membrane protein | 75 | HLA-DR401 | 6 | 1 | 0 |
| | | | HLA-DR402 | 7 | 3 | 0 |
| | | | HLA-DR404 | 6 | 1 | 0 |
| SARS-CoV-2 | Protein 3a | 275 | HLA-DR401 | 22 | 4 | 11 |
| | | | HLA-DR402 | 13 | 4 | 10 |
| | | | HLA-DR404 | 10 | 2 | 0 |
| SARS-CoV-2 | Replicase polyprotein 1a | 4405 | HLA-DR401 | 948 | 228 | 658 |
| | | | HLA-DR402 | 657 | 196 | 409 |
| | | | HLA-DR404 | 865 | 222 | 582 |

*Table 1 continued*

| Virus | Protein | Protein length (# of amino acids) | MHC allele | # of 15mers | # of 9mer cores | # of smoothed 15mers |
|-------|---------|-----------------------------------|------------|-------------|-----------------|----------------------|
| | | | HLA-DR401 | 6 | 1 | 6 |
| | | | HLA-DR402 | 2 | 0 | 0 |
| SARS-CoV-2 | ORF10 protein | 38 | HLA-DR404 | 5 | 1 | 5 |
| | | | HLA-DR401 | 0 | 0 | 0 |
| | | | HLA-DR402 | 0 | 0 | 0 |
| SARS-CoV-2 | Protein non-structural 7b | 43 | HLA-DR404 | 0 | 0 | 0 |
| | | | HLA-DR401 | 8 | 4 | 6 |
| | | | HLA-DR402 | 20 | 5 | 16 |
| SARS-CoV-2 | Uncharacterized protein 14 | 73 | HLA-DR404 | 22 | 4 | 21 |
| | | | HLA-DR401 | 29 | 7 | 27 |
| | | | HLA-DR402 | 35 | 6 | 31 |
| SARS-CoV-2 | Protein 9b | 97 | HLA-DR404 | 37 | 9 | 34 |

in *Figure 3c,d,e*, where individually enriched 15mer sequences are represented as horizontal lines above 15mer stretches in the proteome. Bolded 9mers are identified through register inference as consensus binding cores for these peptides. Only 15mers which contain the bolded 9mer are included in this representation. Non-conserved amino acids within this 9mer are highlighted in yellow.

*Figure 3c* illustrates a region that is not conserved between SARS-CoV-2 and SARS-CoV, where the SARS-CoV-2 peptides containing the core IYQAGSTPC are enriched for binding to all three MHCs, but mutations, including at both P1 and P4 to Proline, discourage binding of the aligned SARS-CoV peptide. *Figure 3e* illustrates a core that is conserved between SARS-CoV and SARS-CoV-2, which can bind only to HLA-DR401, but not to HLA-DR402 or HLA-DR404, likely due to the size of the P1 hydrophobic residue and, for HLA-DR402, the acidic P4 residue. *Figure 3d* illustrates relationships between both viral conservation and MHC preference. In *Figure 3d*, the SARS-CoV peptides containing the core IKNQCVNFN can bind to all three alleles. However, the aligned SARS-CoV-2 peptides containing the core VKNKCVNFN do not bind to HLA-DR401, likely because of the less preferable P1 Valine and basic P4 Lysine, but can bind to HLA-DR402, which prefers these residues. These peptides can bind to HLA-DR404, although only four of the adjacent peptides containing this core enrich, which is below the cutoff of five or more, and since no other adjacent peptides enriched, this would not have been classified as a binder (reflected in *Figure 3b*). This marginal, but below-threshold binding is logical, given that the P4 pocket for HLA-DR404 is similar to HLA-DR401, which does not prefer P4 Lysine, but HLA-DR404 has the same P1 binding pocket as HLA-DR402, which both prefer the P1 Valine in the SARS-CoV-2 peptide.

## Identifying peptide binders missed by computational prediction

Next, we compared our direct experimental assessments with results from computational MHC binding predictions. Prediction algorithms allow for rapid computational screening of potential peptide binders (*Abelin et al., 2019*; *Reynisson et al., 2020*), although they can contain systemic biases (*Rappazzo et al., 2020*). To test the outputs of our direct assessment approach and computational prediction algorithms, we assessed binding of several peptides using a fluorescence polarization competition assay to determine $IC_{50}$ values, as described previously (*Rappazzo et al., 2020*; *Yin and Stern, 2014*). Yeast-formatted peptides (Ala + 15mer + Gly + Gly + Ser) from SARS-CoV-2 spike protein were run through NetMHCIIpan4.0 for binding to HLA-DR401, with binders defined as having ≤10%Rank (Eluted Ligand mode). Yeast display binders to HLA-DR401 were defined via the stringent criteria of (1) enriching in both doped and undoped selections and (2) containing a 9mer that enriched in five or more of the overlapping seven 15mers. 15mers were selected such that they could contain a maximum overlap of eight amino acids with other selected peptides, to avoid selecting

**Table 2.** Peptides selected for fluorescence polarization (FP) experiments for binding to HLA-DR401.
NetMHCIIpan4.0 predictions for HLA-DR401 binding are performed on 15mers plus invariant flanking residues (N-terminal Ala, C-terminal Gly-Gly-Ser) and percent rank values generated using Eluted Ligand mode. Fluorescence polarization is performed on native 15mer peptides without invariant flanking residues.

|  | Spike position | Peptide + flank (A + 15mer + GGS) | NetMHCIIpan4.0 predicted core (A + 15mer + GGS) | NetMHCIIpan4.0 %Rank (A + 15mer + GGS) | 15mer affinity from FP (IC$_{50}$, nM) |
|---|---|---|---|---|---|
| Agreed Binders | 34–48 | ARGVYYPDKVFRSSVLGGS | YYPDKVFRS | 1.49 | 15.8 |
|  | 87–101 | ANDGVYFASTEKSNIIGGS | VYFASTEKS | 4.28 | 2117 |
|  | 303–317 | ALKSFTVEKGIYQTSNGGS | FTVEKGIYQ | 8.41 | 396.9 |
|  | 362–376 | AVADYSVLYNSASFSTGGS | YSVLYNSAS | 8.36 | 113.7 |
|  | 1015–1029 | AAAEIRASANLAATKMGGS | IRASANLAA | 3.13 | 105.4 |
|  | 1112–1126 | APQIITTDNTFVSGNCGGS | ITTDNTFVS | 7.32 | 527.0 |
| Yeast-Enriched Binders | 165–179 | ANCTFEYVSQPFLMDLGGS | YVSQPFLMD | 64.83 | 14,652 |
|  | 172–186 | ASQPFLMDLEGKQGNFGGS | FLMDLEGKQ | 20.34 | 123.2 |
|  | 286–300 | ATDAVDCALDPLSETKGGS | VDCALDPLS | 32.68 | 521.6 |
|  | 373–387 | ASFSTFKCYGVSPTKLGGS | YGVSPTKLG | 16.59 | 18,452 |
|  | 469–483 | ASTEIYQAGSTPCNGVGGS | IYQAGSTPC | 18.22 | 67.7 |
|  | 580–594 | AQTLEILDITPCSFGGGGS | LEILDITPC | 62.00 | 119.9 |
|  | 739–753 | ATMYICGDSTECSNLLGGS | YICGDSTEC | 70.91 | 14.4 |
|  | 920–934 | AQKLIANQFNSAIGKIGGS | FNSAIGKIG | 20.47 | 1121 |
| NetMHC-Predicted Binders | 113–127 | AKTQSLLIVNNATNVVGGS | IVNNATNVV | 8.74 | >50,000 |
|  | 492–506 | ALQSYGFQPTNGVGYQGGS | YGFQPTNGV | 4.11 | 454.7 |
|  | 1151–1165 | AELDKYFKNHTSPDVDGGS | YFKNHTSPD | 5.74 | 35,510 |
| Agreed Non-Binders | 534–548 | AVKNKCVNFNFNGLTGGGS | FNFNGLTGG | 57.13 | >50,000 |
|  | 1079–1093 | APAICHDGKAHFPREGGGS | ICHDGKAHF | 80.47 | >50,000 |

peptides with redundant 9mer cores. While yeast-enriched peptides were largely consistent with computational prediction, we selected sets of sequences which disagreed between computation and experiment, as well as a several sequences that yeast display and NetMHCIIpan4.0 both classified as either binders or non-binders (*Table 2*). A length-matched version of the commonly studied Influenza A Virus HA$_{306-318}$ peptide (APKYVKQNTLKLATG) known to bind HLA-DR401 (*Hennecke and Wiley, 2002*; *Rappazzo et al., 2020*) was also included as a positive control. *Figure 4—figure supplement 1* shows a comparison of yeast-enriched and NetMHCpan4.0 predicted binders, with boxed sequences selected for testing by fluorescence polarization.

The resulting fluorescence polarization IC$_{50}$ data from the native 15mer peptides are shown in *Table 2* and *Figure 4—figure supplement 2*. Peptides which both enriched in yeast display and were predicted by NetMHCIIpan4.0 to bind ('Agreed Binders') all showed IC$_{50}$ values consistent with binding, each with IC$_{50}$ < 2.2 μM. Similarly, peptides which were agreed non-binders showed no affinity for HLA-DR401, with IC$_{50}$ > 50 μM.

All eight 'Yeast-Enriched Binders', which enriched in the yeast display assay but were not predicted to bind via NetMHCIIpan4.0, showed some degree of binding, with IC$_{50}$ values distributed from 14 nM (higher affinity than the HA control peptide) to 18 μM (weak, but measurable, binding). Retrospectively, the weakest two binders appear to be enriching in the yeast display assay using the peptide linker or have a binding core offset from center. Interestingly, NetMHCIIpan4.0 predictions on the peptides identified via yeast display proved highly sensitive to the length or content of the flanking sequences: if we repeat predictions on only the antigen-derived 15mer sequences without the flanking sequences, NetMHCIIpan4.0 recovers four of its former false-negative peptides (*Table 3*; peptides

**Table 3.** Effects of peptide flanking sequences on NetMHCIIpan4.0 predictions for HLA-DR401 binding and measured fluorescence polarization (FP) values for overlapping peptides.

Yeast display-enriched peptides that are predicted to bind by NetMHCIIpan4.0 when without flanking residues, plus offset variants of these peptides, which are not predicted to bind, with or without flanking sequence. Yeast display register-inferred consensus cores are highlighted in bold. NetMHCIIpan4.0 percent rank values are generated using Eluted Ligand mode.

| Spike position | Sequence | NetMHCIIpan4.0 predicted core (A + 15mer + GGS) | NetMHCIIpan4.0 %Rank (A + 15mer + GGS) | NetMHCIIpan4.0 predicted core (15mer) | NetMHCIIpan4.0 %Rank (15mer) | 15mer affinity from FP (IC50, nM) |
|---|---|---|---|---|---|---|
| 172–186 | SQP**FLMDLEGKQ**GNF | FLMDLEGKQ | 20.34 | FLMDLEGKQ | 4.1 | 123.2 |
| 173–187 | QP**FLMDLEGKQ**GNFK | FLMDLEGKQ | 27.73 | FLMDLEGKQ | 12.21 | 8613 |
| 286–300 | TDA**VDCALDPLS**ETK | VDCALDPLS | 32.68 | VDCALDPLS | 9.8 | 1154 |
| 287–301 | DA**VDCALDPLS**ETKC | VDCALDPLS | 42.42 | VDCALDPLS | 22.57 | 4,393 |
| 469–483 | STE**IYQAGSTPC**NGV | IYQAGSTPC | 18.22 | IYQAGSTPC | 5.41 | 67.7 |
| 467–481 | DISTE**IYQAGSTPC**N | IYQAGSTPC | 11.47 | IYQAGSTPC | 12.61 | 4875 |
| 471–485 | E**IYQAGSTPC**NGVEG | YQAGSTPCN | 39.17 | YQAGSTPCN | 21.81 | 12,519 |
| 920–934 | QKL**IANQFNSAI**GKI | FNSAIGKIG | 20.47 | IANQFNSAI | 7.89 | 1495 |
| 921–935 | KL**IANQFNSAI**GKIQ | FNSAIGKIQ | 18.3 | IANQFNSAI | 19.79 | 11,937 |

listed at the top in each section of the table). We will refer to these four peptides as 'flank-sensitive centered peptides', as they each have the consensus 9mer core centered in the peptide.

To further investigate the relationship with flanking residues, we selected five additional peptides ('offset peptides') matching three criteria; these offset peptides were (1) enriched in the yeast display assay, (2) share an overlapping core with the four flank-sensitive centered peptides, but are (3) not predicted by NetMHCIIpan4.0 to be binders (either with or without invariant flanking sequence added). All five offset peptides have their predicted cores offset by one to two amino acids from center, leaving at minimum one amino acid on both ends of the 9mer core for each peptide. All five offset peptides exhibit some binding, with $IC_{50}$ values below 13 μM. Each peptide is lower affinity than its overlapping centered counterpart, illustrating effects of flanking residues on peptide binding, although some over-estimation of these effects in NetMHCIIpan4.0 predictions are present.

We tested three 'NetMHC-Predicted Binders', which were predicted to bind by NetMHCIIpan4.0, but were not enriched (nor did any neighboring sequences within an offset of four amino acids) in the yeast display assay (*Table 2*). Of these, one bound to HLA-DR401 ($IC_{50}$ 475 nM), while two showed minimal binding with IC50 > 35 μM, which is above the maximum 20 μM concentration tested. All three were predicted by NetMHCIIpan4.0 to bind with or without the invariant flanking sequences (Eluted Ligand mode %Rank: 5.7, 4.1, 8.7 (with flanking residues) and 2.3, 0.6, 7.0 (without flanking residues), for ELDKYFKNHTSPDVD, LQSYGFQPTNGVGYQ, and KTQSLLIVNNATNVV, respectively).

Of the eight 'Yeast-Enriched Binders' in *Table 2*, six contain cysteine residues, which have been shown to be systematically absent from other datasets, including those from monoallelic mass spectrometry (*Abelin et al., 2019*; *Barra et al., 2018*), yet present in yeast display-derived datasets (*Rappazzo et al., 2020*). To test for non-specific binding due to cysteine, two cysteine-containing 'Agreed Non-Binders' were also tested and showed no affinity for HLA-DR401, suggesting that cysteine itself is not causing non-specific binding. In the fluorescence polarization dataset, the highest affinity binder (14 nM) contained cysteine and was missed by NetMHCIIpan4.0 predictions (Eluted Ligand mode %Rank: 71 [with flanking residues] and 28 [without flanking residues]).

The relationship between measured $IC_{50}$ values and NetMHCIIpan4.0 predicted values for all 15mer SARS-CoV-2 spike peptides tested is shown in *Figure 4*, *Figure 4—figure supplement 2*, *Figure 4—figure supplement 3*.

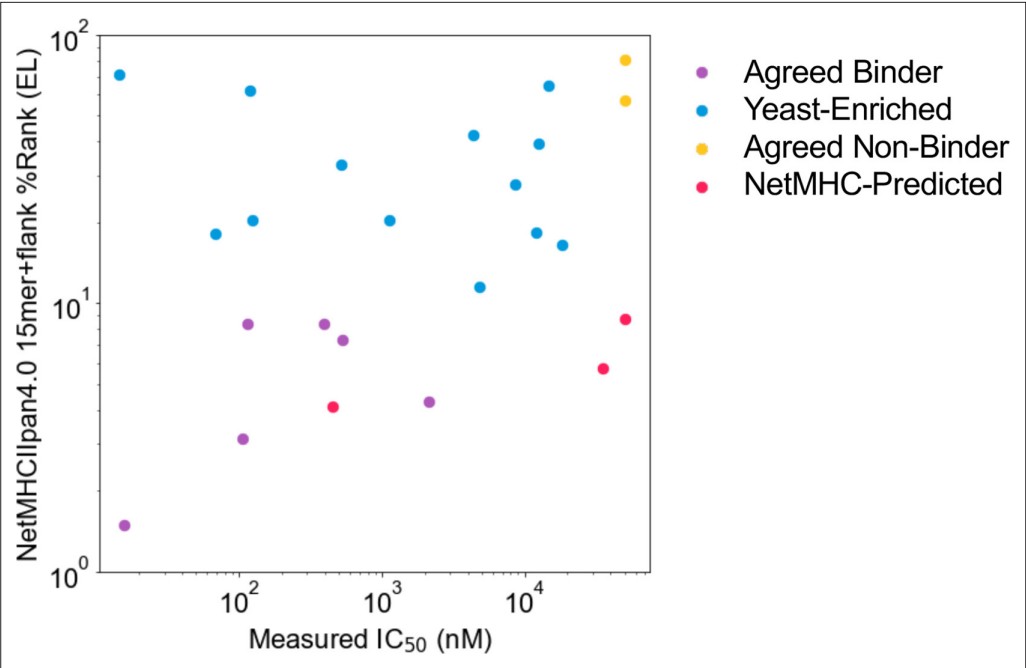

**Figure 4.** Comparing measured $IC_{50}$ values and computational prediction. Relationship between measured $IC_{50}$ values and NetMHCIIpan4.0 predicted ranks in Eluted Ligand mode (EL) on invariant-flanked sequences. Data points are colored by label, and $IC_{50}$ values $\geq$ 50 μM are set to 50 μM.

The online version of this article includes the following source data and figure supplement(s) for figure 4:

**Figure supplement 1.** Comparing defined library selection with algorithmic predictions: SARS-CoV-2 spike protein.

**Figure supplement 2.** Titration curves for peptides tested via fluorescence polarization for binding to HLA-DR401, by category.

**Figure supplement 2—source data 1.** Fluorescence polarization measurements for peptide-HLA-DR401 binding.

**Figure supplement 3.** Comparing measured $IC_{50}$ values and prediction.

## Comparing whole dengue serotype proteomes for common MHC-binding peptides

Defined yeast display libraries can generate data for diverse objectives. Dengue viruses typically cause most severe disease after a second infection with a serotype different from the first infection, due to antibody-dependent enhancement (*Guzman et al., 2016*), which makes T cell-directed therapeutics a potentially attractive means of combating disease. To profile and compare MHC binding across serotypes, we generated libraries containing 12,672 dengue-derived peptides, covering the entire proteomes of dengue serotypes 1–4. These libraries were on HLA-DR401 and HLA-DR402 and had coverage of 98% and 96% of the dengue library members after construction, respectively (*Supplementary file 1c*).

Peptides from homologous regions of the four dengue serotypes have different MHC binding ability, as illustrated in *Figure 5a* for binding to HLA-DR401. The proteins encoded in the dengue genome are indicated along the horizontal axis (C: capsid; M: membrane; E: envelope; NS: nonstructural proteins). Peptides that enriched in the yeast display assay are marked by a line (serotype 1 in blue, serotype 2 in purple, serotype 3 in red, and serotype 4 in gray). The proteome is smoothed to nine amino acid stretches (as in *Figure 3b*), with a given nine amino acid region marked as a hit if five or more of the seven adjacent peptides enrich. For each 9mer, the maximum number of serotypes with a conserved identical 9mer at that position is indicated at the top in black.

These data can reveal relationships between conservation and binding ability. *Figure 5b–d* shows enrichment data for individual 15mer peptides, with consensus inferred 9mer cores in bold and non-conserved amino acids in these cores highlighted in yellow, as in *Figure 3c–e*. Conserved cores which

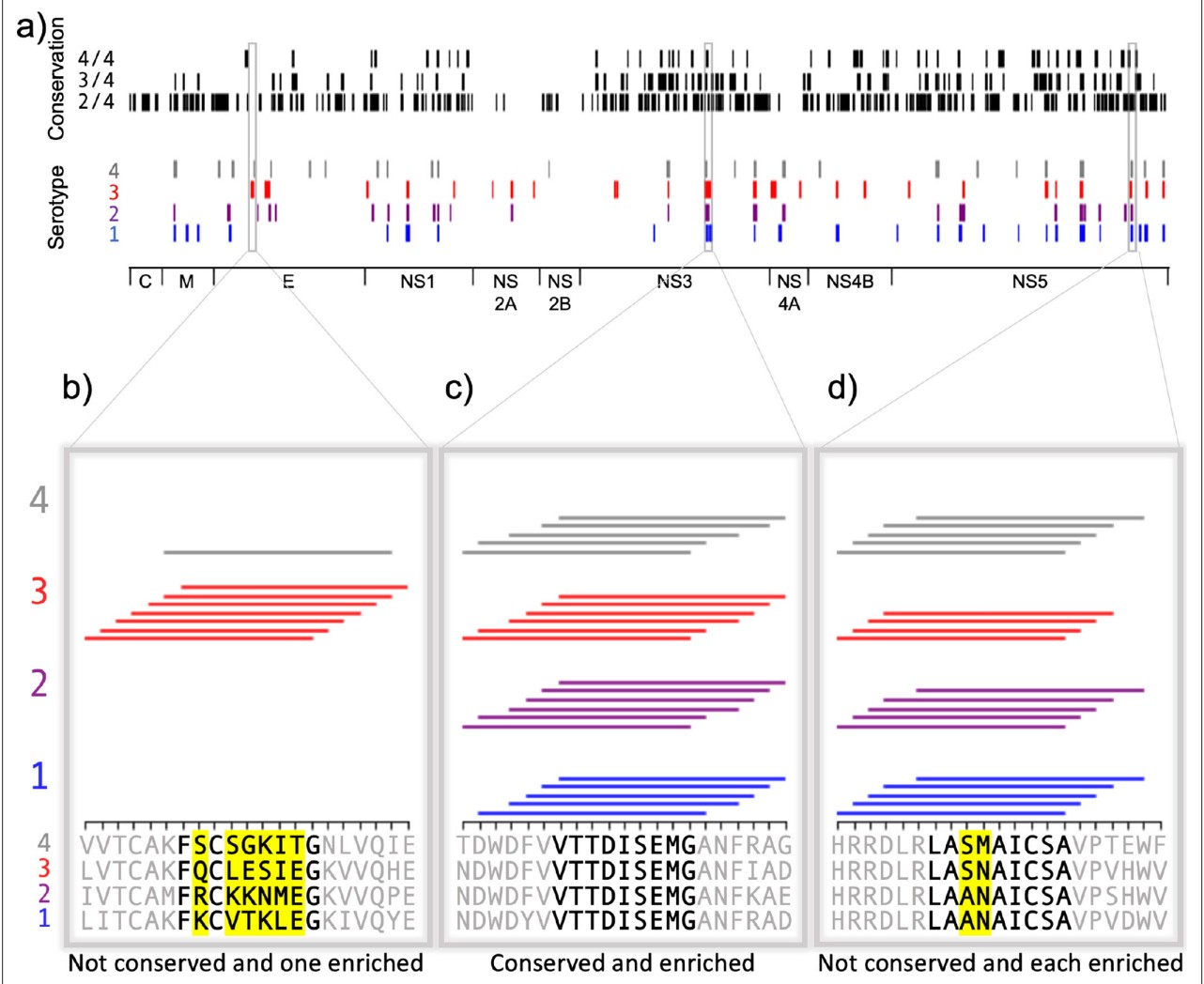

**Figure 5.** Conservation and enrichment of dengue virus serotypes 1–4. (**a**) Conservation and enrichment of 9mer peptides along four aligned dengue serotypes. All stretches of nine amino acids are compared across the four serotypes and conservation is indicated with a black vertical line (i.e. 2, 3, or 4 of four serotypes conserved). 9mers which enriched on HLA-DR401 are also indicated, colored by virus serotype. (**b–d**) Zoomed regions, showing enrichment for individual 15mer peptides to HLA-DR401. Only peptides which contain the bolded 9mer sequence are shown. Amino acids in the bolded 9mer that are not conserved between serotypes are highlighted in yellow. Insets show regions which are differently conserved and enriched: (b) non-conserved sequences with peptides from one serotype enriched; (c) conserved sequences enriched across all serotypes; (d) non-conserved sequences which are enriched.

show binding ability (*Figure 5c*) may be ideal T cell targets. However, the permissiveness of the binding groove allows for peptides to bind that have mutations at the anchors, such as in NS5 (*Figure 5d*), where P4 Asn and P4 Met both allow binding. Interestingly, the serotype 3 core (LASNAICSA) only enriched in four peptides, which is below our described cutoff for high-confidence peptide cores. However, three adjacent peptides enriched and register inference for these peptides identifies the non-native, linker-containing version of the LASNAICSA core as binding in the MHC-binding groove. This results in an adjacent 9mer being highlighted as a binder in this region (*Figure 5a*) because overlapping 15mers enrich in five or more of the seven adjacent peptides. With this in mind, care must be taken for core identification in enriched regions and can be aided by coupling enrichment with register inference of enriched peptides. Further, we can also see relationships between conservation and binding in non-conserved regions, such as in the envelope protein (*Figure 5b*) with the mutations in serotype 3 enabling binding.

## Discussion

CD4[+] T cell responses play important roles in infection, autoimmunity, and cancer. By extension, understanding peptide-MHC binding is critical for identifying and engineering T cell epitopes. Here, we present an approach to directly assess defined libraries of peptides covering whole pathogen proteomes for binding to MHC-II proteins. We examine alternative modes of selection and utilize overlapping peptides to determine high-confidence binders. We demonstrate the utility of this approach by identifying binders that are missed by prediction algorithms, highlighting a prediction algorithm bias against cysteine-containing peptides and sensitivity to peptide flanking residues (*Table 2* and *Table 3*). Finally, this approach can be utilized for different objectives, including comparing binding to multiple MHC alleles (*Figure 3*) or comparing peptides from related pathogen sequences for MHC-II binding (*Figure 5*). Whole protein- or proteome-scale analysis across related viruses provides insight into relationships between conserved epitopes and MHC binding (*Figures 3b and 5a*) and specific examples validate the consistency with the underlying biophysics of peptide-MHC binding (*Figures 3c–e , and 5b–d*).

When compared to previously described yeast display approaches to identify peptides binding to MHC-II molecules, our approach benefits from recent advances in next-generation sequencing and pooled oligonucleotide synthesis. Other library generation methods for peptide-MHC-II binding have relied on synthetic peptides (*Liu et al., 2021a*), randomized, DNA-encoded peptides (*Rappazzo et al., 2020*), or digested DNA from amplified viral genomes (*Wen et al., 2008*), which are impractical for comprehensive assessment of defined proteome-scale libraries. Next-generation sequencing further enables a comprehensive and granular view of peptide enrichment, beyond sequencing and validation of individual clones (*Wen et al., 2008*).

This approach for direct assessment shows benefit compared to prediction algorithms for identifying binders, particularly for finding weak peptide binders. Weak binding peptides have been reported to be less immunodominant than strong MHC-binding peptides (*Burger et al., 2021*; *Lazarski et al., 2005*; *Wu et al., 2019*); however, there are also examples of weak MHC-binding peptides which can elicit T cell responses and be of clinical relevance in disease contexts, including in autoimmunity and cancer (*Latek et al., 2000*; *Levisetti et al., 2008*; *Valmori et al., 1998*; *Zarour et al., 2000*). As such, information about weak binding peptides can be of potential scientific and clinical relevance.

The overlapping peptides in our library were useful for identifying enriched cores, especially when combined with our register inference to identify consensus cores shared between these overlapping peptides. NetMHCIIpan4.0 exhibits a sensitivity to length and register, which may cause users to miss binders, albeit potentially of lower affinity. Of the overlapping peptides we tested to study this phenomenon, NetMHCIIpan4.0 correctly ranked the affinities of the overlapping peptides (*Table 3*), but missed binders. *Figure 4—figure supplement 1* also highlights the sensitivity of NetMHCIIpan4.0 to flanking sequences, where neighboring peptides with shared cores often are not predicted to bind, resulting in fewer clusters of peptides. Comparison with yeast display datasets also highlights several non-binding peptides predicted by NetMHCIIpan4.0 to be binders. Coupling this yeast display approach with computational predictions can be useful for identifying false positive predicted peptides in order to correctly prioritize peptides of interest.

Our work reveals insights on the design of epitope identification experiments, including the utility of overlapping peptides and considerations for comparing libraries of unbiased and proteome-derived peptides. Design of defined libraries with sources of redundancy, such as overlapping peptides, was critical for determining binders with higher degrees of confidence and allowed us to apply stringent cutoffs for individual peptides. Overlapping peptides allowed us to account for construct-specific confounding effects, such as the peptides binding using non-native residues in the linker. Future iterations can change the sequence of the linker, such as defining favorable P(–1) and P10 anchors to fix the register (*Rappazzo et al., 2020*), although these adaptations would likely require MHC-specific knowledge in advance and may need to be altered for different MHCs. Additionally, the engineered redundancy and multiple modes of selection result in hyperparameters that can be tuned to meet users' stringency requirements, such as defining different thresholds for calling individual 15mer binders or alternative integration of overlapping binders. Additionally, our comparison of unbiased and proteome-derived libraries highlights how aggregate motifs may be affected by underlying amino acid preferences found in protein sequences themselves, which may inadvertently disfavor

sequences that can bind strongly to MHC molecules yet consist of amino acid covariates that are not as commonly found in proteins.

Further, this approach can be used to study MHC binding between similar viruses, as done with the dengue proteomes and the spike proteins from SARS-CoV-2 and SARS-CoV, highlighting regions where mutations disrupt binding as well as regions where binding is unperturbed. This method can also be rapidly adapted to study future sequences if pathogens evolve over time.

As experimental approaches and computational approaches continue to co-develop, they present complementary benefits. Though this platform allows for rapid assessment of peptide-MHC binding, the speed of computational prediction surpasses experimental approaches. NetMHCIIpan4.0 prediction and yeast display selections identified sets of non-overlapping misses, highlighting a utility for both. Additionally, all agreed binders and non-binders matched fluorescence polarization results, suggesting a consensus of yeast display enrichment and algorithmic prediction provide high-confidence results. Approaches such as yeast display assessment can be used to complement computational approaches, such as for identifying cysteine-containing peptides which are still under-predicted by algorithms. Similarly, prediction algorithms can be trained using large, quality datasets to account for biases. Training sets can specifically be augmented with data from defined peptide libraries designed to study peptides where current algorithmic predictions are of lower confidence. In another application, our platform to assess peptide-MHC binding can be used to design high-throughput assays to test peptide immunogenicity in clinical samples (*Klinger et al., 2015*; *Snyder et al., 2020*).

Defined yeast display peptide libraries can also be readily applied to identification of T cell ligands and present an opportunity for identifying unknown ligands from orphan TCRs known to respond to a proteome of interest (*Birnbaum et al., 2014*; *Gee et al., 2018*). Yeast have also been previously utilized as artificial antigen-presenting cells to stimulate T cell hybridomas (*Wen et al., 2008*), making it possible to further streamline antigen discovery efforts. As DNA synthesis and sequencing continue to advance, defined peptide libraries expanding beyond viral proteomes to covering whole bacterial or human proteomes will be possible, and could present opportunities for investigating autoimmune diseases, which frequently have strong MHC-II associations (*Karnes et al., 2017*). Such tools would be rich resources for identifying both peptide-MHC binders and TCR ligands.

# Methods

**Key resources table**

| Reagent type (species) or resource | Designation | Source or reference | Identifiers | Additional information |
|---|---|---|---|---|
| Strain, strain background (*Saccharomyces cerevisiae*) | RJY100 | PMID:26333274 | | |
| Cell line (*Trichoplusia ni*) | High Five cells | Thermo Fisher | Cat#:B85502 | |
| Cell line (*Spodoptera frugiperda*) | Sf9 cells | Thermo Fisher | Cat#:11496015 | |
| Antibody | Anti-Myc-AlexaFluor647 (Mouse monoclonal) | Cell Signaling Technology | Cat#:2233 | Library selections: 1:100 |
| Recombinant DNA reagent | Peptide-MHC-II with cleavable peptide linker in pYal (plasmid) | PMID:32887877 | | |
| Recombinant DNA reagent | HLA-DR401 in pAcGP67a (plasmid) | PMID:32887877 | | |
| Recombinant DNA reagent | HLA-DM in pAcGP67a (plasmid) | PMID:32887877 | | |
| Peptide, recombinant protein | 3C protease | Other | | Purified from *Escherichia coli* BL21 cells |
| Software, algorithm | PandaSeq | PMID:22333067 | | |
| Software, algorithm | NetMHCIIpan4.0 | PMID:32406916 | | |
| Software, algorithm | Peptide register inference algorithm | This paper | | See Code Availability |
| Software, algorithm | Prism | GraphPad Prism software (http://www.graphpad.com/) | | Version: 9.3 |

## Library design and creation

Yeast display libraries were designed to cover all 15mer sequences within a given proteome, with step size 1. Reference proteomes used in creating defined libraries were accessed from Uniprot, with the following Proteome IDs. SARS-CoV-2: UP000464024, SARS-CoV: UP000000354, dengue serotype 1: UP000002500, dengue serotype 2: UP000180751, dengue serotype 3: UP000007200, dengue serotype 4: UP000000275. The dengue proteome is expressed as a single polypeptide, and peptides were generated from that contiguous stretch.

Each library peptide is encoded in DNA space, with specific codons selected randomly from possible codons, with probabilities matching yeast codon usage (GenScript Codon Usage Frequency Table). The DNA-encoded peptide sequences were flanked by invariant sequences from the yeast construct for handles in amplification and cloning, and the DNA oligonucleotide sequences were ordered from Twist Bioscience (South San Francisco, CA), with maximum length of 120 nucleotides. The DNA oligo pool was amplified in low cycle PCR, followed by amplification with construct DNA using overlap extension PCR. This extended product was assembled in yeast with linearized pYal vector at a 5:1 insert:vector via electroporation with electrocompetent *Saccharomyces cerevisiae* RJY100 yeast (*Van Deventer et al., 2015*). Primers utilized in this study are included in *Supplementary file 1d*.

HLA-DR401 and HLA-DR402 libraries were generated using previously described vectors (*Rappazzo et al., 2020*) which contain mutations from wild type Metα36Leu, Valα132Met, Hisβ33Asn, and Aspβ43Glu to enable proper folding without disrupting TCR or peptide contact residues (*Rappazzo et al., 2020*). HLA-DR404 was generated using the same stabilizing mutations. As previously described (*Rappazzo et al., 2020*), the peptide C-terminus is connected to the MHC construct via a Gly-Ser linker (*Figure 1a*), and the N-terminus of the peptide includes an extra alanine to ensure consistent cleavage between the construct and its signal peptide.

The previously described null library (*Dai et al., 2021*) was generated with a peptide encoded as 'NNNTAANNNNNNNNNTAGNNNNNNNNNNNNNNTGANNNNNN', where 'N' indicates any nucleotide and encodes 10 random amino acids and three stop codons. This library was similarly generated in yeast using electrocompetent RJY100 yeast.

## Peptide visualizations and predictions

Data visualizations of viral conservation and enrichment were generated using custom scripts. For each 9mer stretch in a protein of interest, there are seven 15mer sequences that overlap and contain that 9mer. We calculate how many of these seven 15mers enriched in both the doped and undoped libraries. If five or more of the seven 15mers enriched, that stretch is marked as a 'hit'. To examine conservation between viruses, viral proteins are aligned using ClustalOmega (*Madeira et al., 2019*). Aligned 9mer stretches are compared between viruses and identical stretches are considered conserved. Hits are determined individually for each virus before merging, such that gaps in sequence alignments do not affect calculations of enrichment for a given virus.

Representations of 15mer hits (as in *Figure 3*, *Figure 5* and *Figure 4—figure supplement 1*) were generated using in-house scripts, such that a 15mer that enriched in both the doped and undoped library was marked as a horizontal line above the relevant 15mer sequence. Only 15mers containing the bolded 9mer in *Figure 3* and *Figure 5* were included.

NetMHCIIpan4.0 webserver was used for computational predictions (*Reynisson et al., 2020*), where a binder is defined as having a predicted percent rank ≤10%, as defined in the webserver instructions.

## Yeast library selections

Library selections were consistent with previous peptide-MHC-II yeast display dissociation studies (*Dai et al., 2021*; *Rappazzo et al., 2020*). Yeast were washed into pH 7.2 PBS with 1 µM 3C protease and incubated at room temperature for 45 min. Yeast were then washed into 4°C acid saline (150 mM NaCl, 20 mM citric acid, pH 5) with 1 µM HLA-DM and incubated at 4°C overnight. Each step takes place in the presence of competitor peptide (HLA-DR401: HA$_{306-318}$ PKYVKQNTLKLAT, 1 µM; HLA-DR402: CD48$_{36-51}$ FDQKIVEWDSRKSKYF, 5 µM; HLA-DR404: NKVKSLRILNTRRKL, 5 µM *Vita et al., 2019*). Non-specific binders are removed by incubating yeast with anti-AlexaFluor647 magnetic beads and flowed over a magnetic Milltenyi column at 4°C. A positive selection follows, comprised of incubation with anti-Myc-AlexaFluor647 antibody (1:100 volume:volume; Myc tag (9B11) Mouse mAb

AlexaFluor647 Conjugate #2233 Cell Signaling Technology) and anti-AlexaFluor647 magnetic beads (1:10 volume:volume) and flowed over a Milltenyi column on a magnet at 4°C, such that yeast with bound peptide are retained on the column. These yeast are eluted, grown to confluence in at 30°C in SDCAA media (pH 5), and sub-cultured in at 20°C SGCAA media (pH 5) at OD600=1 for 2 days. The first round of selections of doped libraries were conducted on 180 million yeast (SARS-CoV-2 library) or 400 million yeast (dengue library) to ensure at least 20-fold coverage of peptides. Subsequent rounds of doped library selection, and all rounds of undoped library selections, were performed on 20–25 million yeast. Before each round of selection, a sample of yeast are stained with an anti-Myc antibody to check induction of protein expression, as in *Figure 1c*.

## Library sequencing and analysis

Libraries were deep sequenced to determine their composition after each round of selection. Plasmid DNA was extracted from 10 million yeast from each round of selection using the Zymoprep Yeast Miniprep Kit (Zymo Research), following the manufacturer's instructions. Amplicons were generated through PCR, covering the peptide sequence through the 3C cut site. A second PCR round was performed to add i5 and i7 sequencing handles and in-line index barcodes unique to each round of selection. Amplicons were sequenced on an Illumina MiSeq using paired-end MiSeq v2 300 bp kits at the MIT BioMicroCenter.

Paired-end reads were assembled using PandaSeq (*Masella et al., 2012*). Peptide sequences were extracted by identifying correctly encoded flanking regions, and were filtered to ensure they matched designed members of the library or the randomized null construct encoding, providing a stringent threshold for contamination and PCR and read errors.

The resulting data are analyzed for convergence, as described in the main text. Once a library has converged, it is likely that changes in subsequent rounds of selection are due to stochastic variation rather than improved binding.

Peptide and read count data are in *Supplementary file 1a and c*. Columns are labelled as: [count]_ [doped or undoped library]-[HLA-DR allele] post-R[round number]-[positive or negative, for undoped libraries]. For unselected 'R0' libraries, an additional suffix may be present to indicate whether the library was sequenced before or after doping into the null library. Also indicated are: amino acid sequence ('aa'), encoding DNA sequence ('dna'), and whether the sequence matched the null library (indicated in 'doped_match' or 'name').

## Register inference and sequence logos

The 9mer core of enriched sequences was inferred using an in-house alignment algorithm. In this approach, we utilize a 9mer PWM, which we assess at different offsets along the peptide. We one-hot encode sequences and pad with zeros on the C-terminus of the peptide; to assess seven native registers and four non-native registers, we pad the peptides with four zeros. Three of the non-native registers utilize the linker at the P9 anchor but not the P6 anchor, and the addition of a fourth register captures a minority set of peptides which utilize Gly-Gly-Ser-Gly of the linker at P6 through P9 in the groove. Register-setting is performed with zero-padded 15mers, rather than 15mers flanked by invariant flanking residues, because the PWM would otherwise align all sequences to the invariant region.

At the start, we randomly assign peptides to registers and generate a 9mer PWM. Over subsequent iterations, peptides are assigned to new registers and the PWM was updated. Assignments are random but biased, such that clusters corresponding to registers that match the PWM are favored. Specifically, at each assignment we first take out the sequence under consideration from the PWM. The PWM then defines an energy value for each register shift of a given peptide, which is then used to generate a Boltzmann distribution from which we sample the updated register shift. The stochasticity is decreased over time by raising the inverse temperature linearly from 0.05 to 1 over 60 iterations, simulating 'cooling' (*Andreatta et al., 2017*). A final deterministic iteration was carried out, where the distribution concentrates entirely on the optimal register shift.

After register inference, sequence logo visualizations of the 9mer cores were generated using Seq2Logo-2.0 with default settings, except using background frequencies from the SARS-CoV-2 proteome and SARS-CoV spike and nucleocapsid proteins (*Thomsen and Nielsen, 2012*). For registers with the C-terminus utilizing the C-terminal linker, the relevant linker sequence was added to

achieve a full 9mer sequence for visualizing the full 9mer core. For HLA-DR401, distribution among registers, starting from N-terminally to C-terminally aligned in the peptide, is: 161, 237, 227, 238, 231, 279, 237, 266, 271, 202, 118.

## Recombinant protein expression

HLA-DM and HLA-DR401 were expressed recombinantly in High Five insect cells (species *Trichoplusia ni*; Thermo Fisher B85502) using a baculovirus expression system, as previously described (**Birnbaum et al., 2014**; **Rappazzo et al., 2020**). Ectodomain sequences of each chain were formatted with a C-terminal poly-histidine purification tag and cloned into pAcGP67a vectors. Each vector was individually transfected into Sf9 insect cells (species *Spodoptera frugiperda*; Thermo Fisher 11496015) with BestBac 2.0 linearized baculovirus DNA (Expression Systems; Davis, CA) and Cellfectin II Reagent (Thermo Fisher), and propagated to high titer. Viruses were co-titrated for optimal expression to maximize balanced MHC heterodimer formation, co-transduced into High Five cells, and grown for 48–72 hr at 27°C. The secreted protein was purified from pre-conditioned media supernatant with Ni-NTA resin and purified via size exclusion chromatography with an S200 increase column on an AKTA PURE FPLC (GE Healthcare). To improve protein yields, the HLA-DRB1*04:01 chain was expressed with a CLIP$_{87-101}$ peptide (PVSKMRMATPLLMQA) connected to the N-terminus of the MHC chain via a flexible, 3C protease-cleavable linker.

## Fluorescence polarization experiments for peptide IC$_{50}$ determination

Peptide IC$_{50}$ values were determined following a protocol modified from **Yin and Stern, 2014**, as in **Rappazzo et al., 2020**. In the assay, recombinantly expressed HLA-DR401 is incubated with fluorescently labelled modified HA$_{306-318}$ (APRFV{Lys(5,6 FAM)}QNTLRLATG) peptide and a titration series for each unlabelled competitor peptide is added (1.28 nM to 20 µM). A change in polarization value resulting from displacement of fluorescent peptide from the binding groove is used to determine IC$_{50}$ values.

Relative binding at each concentration is calculated as (FP$_{sample}$ – FP$_{free}$)/(FP$_{no\_comp}$ – FP$_{free}$). Here, FP$_{free}$ is the polarization value for the fluorescent peptide alone with no added MHC, FP$_{no\_comp}$ is polarization value for MHC with no competitor peptide added, and FP$_{sample}$ is the polarization value with both MHC and competitor peptide added. Relative binding curves were then generated and fit in Prism 9.3 to the equation y=1/(1+[pep]/IC$_{50}$), where [pep] is the concentration of unlabelled competitor peptide, in order to determine the concentration of half-maximal inhibition, the IC$_{50}$ value.

Each assay was performed at 200 µL, with 100 nM recombinant MHC, 25 nM fluorescent peptide, and competitor peptide (GenScript). This mixture co-incubates in pH 5 binding buffer at 37°C for 72 hr in black flat bottom 96-well plates. Competitor peptide concentrations ranged from 1.28 nM to 20 µM, as a fivefold dilution series. Three replicates are performed for each peptide concentration. Fluorescent peptide-only, no competitor peptide, and binding buffer controls were also included. Our MHC was expressed with a linked CLIP peptide, so prior to co-incubation, the peptide linker is cleaved by addition of 3C protease at 1:10 molar ratio at room temperature for 1 hr; the residual cleaved 100 nM CLIP peptide is not expected to alter peptide-binding measurements.

Measurements were taken on a Molecular Devices SpectraMax M5 instrument. G-value was 1.1 for each plate, as calculated per the manufacturer's instructions for each plate based on fluorescent peptide-only wells minus buffer blank wells, with 35 mP reference for 5,6FAM (Fluorescein setting). Measurements were made with 470 nm excitation and 520 nm emission, 10 flashes per read, and default PMT gain high.

## Cell lines

High Five insect cells (species *T. ni*; Thermo Fisher B85502) and Sf9 insect cells (species *S. frugiperda*; Thermo Fisher 11496015) were utilized for recombinant protein production. Their identities have been confirmed functionally through characterization of their secreted baculovirus and protein production.

RJY100 (*S. cerevisiae*) (**Van Deventer et al., 2015**) were utilized for yeast surface display. Yeast identity has been confirmed through analysis of growth and library generation characteristics and engineered protein surface expression profiling.

Sf9 cells have tested mycoplasma negative, and High Five cells have not been independently tested since purchased from the manufacturer. RJY100 yeast have not been mycoplasma tested.

## Code availability

Scripts used for data processing and visualization are publicly available at https://github.com/birn-baumlab/Huisman-et-al-2022, copy archived at swh:1:rev:694c6976275bb02d1d498d0e8a01523a1c b1799d; *Huisman, 2022*.

## Acknowledgements

We would like to thank Patrick Holec for feedback on this manuscript, and the MIT BioMicro Center for library sequencing. This work was supported in part by the Koch Institute Support (core) Grant P30-CA14051 from the National Cancer Institute. This work was supported by National Institute of Health (U19-AI110495), the Packard Foundation to MEB, a Schmidt Futures grant to DKG and MEB, and a National Science Foundation Graduate Research Fellowship to BDH.

## Additional information

### Competing interests

David K Gifford: is a founder of ThinkTx. Michael E Birnbaum: is an equity holder in 3T Biosciences, and is a co-founder, equity holder, and consultant of Kelonia Therapeutics and Abata Therapeutics. The other authors declare that no competing interests exist.

### Funding

| Funder | Grant reference number | Author |
| --- | --- | --- |
| David and Lucile Packard Foundation | | Michael E Birnbaum |
| Schmidt Futures | | David K Gifford Michael E Birnbaum |
| National Science Foundation | | Brooke D Huisman |
| National Institutes of Health | U19-AI110495 | Michael E Birnbaum |

The funders had no role in study design, data collection and interpretation, or the decision to submit the work for publication.

### Author contributions

Brooke D Huisman, Conceptualization, Data curation, Formal analysis, Investigation, Methodology, Software, Validation, Visualization, Writing – original draft, Writing – review and editing; Zheng Dai, Formal analysis, Software, Writing – review and editing; David K Gifford, Funding acquisition, Project administration, Supervision, Writing – review and editing; Michael E Birnbaum, Conceptualization, Funding acquisition, Project administration, Supervision, Writing – original draft, Writing – review and editing

### Author ORCIDs

Brooke D Huisman http://orcid.org/0000-0002-6229-6498
Michael E Birnbaum http://orcid.org/0000-0002-2281-3518

### Decision letter and Author response

Decision letter https://doi.org/10.7554/eLife.78589.sa1
Author response https://doi.org/10.7554/eLife.78589.sa2

## Additional files

### Supplementary files

• Supplementary file 1. Data from generation and deep sequencing of yeast-displayed peptide-major histocompatibility complex (MHC) libraries. (a) Peptide count data from deep sequencing of

the SARS-CoV and SARS-CoV-2 libraries. (b) Inferred binding registers of enriched peptides from the SARS-CoV and SARS-CoV-2 libraries. (c) Peptide count data from deep sequencing of the dengue libraries. (d) Primer sequences utilized for generation and deep sequencing of yeast display libraries.

• MDAR checklist

## Data availability

All deep sequencing data are deposited on the Sequence Read Archive (SRA), with accession codes PRJNA806475 [https://www.ncbi.nlm.nih.gov/bioproject/PRJNA806475] and PRJNA708266 [https://www.ncbi.nlm.nih.gov/bioproject/PRJNA708266]. Processed peptide data are available as a supplemental dataset, and source data for fluorescence polarization measurements are provided.

The following dataset was generated:

| Author(s) | Year | Dataset title | Dataset URL | Database and Identifier |
|---|---|---|---|---|
| Huisman BD, Birnbaum ME | 2022 | High-throughput yeast display approach to profiling pathogen proteomes for MHCII binding | https://www.ncbi.nlm.nih.gov/bioproject/?term=PRJNA806475 | NCBI BioProject, PRJNA806475 |

The following previously published dataset was used:

| Author(s) | Year | Dataset title | Dataset URL | Database and Identifier |
|---|---|---|---|---|
| Huisman BD, Dai Z, Gifford DK, Birnbaum ME | 2021 | Machine learning optimization of peptides for presentation by class II MHCs | https://www.ncbi.nlm.nih.gov/bioproject/?term=PRJNA708266 | NCBI BioProject, PRJNA708266 |

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
