## [Editor Report]

This paper will be of interest to immunologists who study T cell recognition and informaticians who predict peptide ligands for MHC. It rigorously innovates a high-throughput approach to identify MHC ligands of pathogens on the proteome level using a previously developed yeast display-based platform, including ligands not identified by computational prediction. The data support the conclusions well and open a direction for future work to address the immunological significance of the findings.

---

## [Decision Letter]

**Decision letter after peer review:**

Thank you for submitting your article "A high-throughput yeast display approach to profile pathogen proteomes for MHC-II binding" for consideration by *eLife*. Your article has been reviewed by 2 peer reviewers, and the evaluation has been overseen by a Reviewing Editor and Satyajit Rath as the Senior Editor. The following individual involved in the review of your submission has agreed to reveal their identity: Evan W Newell (Reviewer #1); Leonard Moise (Reviewer #2)

Essential revisions:

1) The authors should reference and discuss technical differences from the approach previously published by Wen et al., J. Immunological Methods, 2008 which also uses yeast display to identify MHC class II binding peptides derived from the influenza virus genome.

2) A discussion about the significance of the observation that the yeast display-based approach identifies MHC II ligands that are not found by NetMHCIIpan4.0 would enhance the paper. This is an important finding, on the one hand, because the method may provide new training data that will improve computational prediction accuracy. On the other hand, many of these sequences are low-affinity binders and may not be immunoreactive as peptide affinity drives T cell response (e.g., PMID: 16039577, PMID: 31253788). How this fits in the context of the oft-heard criticism that computational approaches overpredict would benefit the discussion, as well.

3) Related to the observation referenced above in (2), the authors imply in parts (Abstract, Introduction) that the yeast display method is superior to computational predictions because it identifies MHC II ligands not discovered by computational algorithms. However, the current study is limited to three MHC II alleles, examines only one predictor, and does not provide evidence of T cell validation (nor even discussion) of the SARS-CoV-2 and dengue datasets in the context of published predictions, MHC II binding data, and immunological studies. The balanced approach taken in the Discussion where experimental and computational approaches are said to complement each other is constructive as it recognizes that both methods have advantages and disadvantages and is a good model for portraying their relationship in earlier parts of the paper.

4) There is also one issue of the presentation that requires correction. In Figure 1c, the darker shade of blue in the Round 4 flask in the doped library that represents an increase in enrichment from Round 3 to Round 4 is not matched by the decrease in myc staining in Round 4 relative to Round 3.

---

## [Author Response]

Essential revisions:1) The authors should reference and discuss technical differences from the approach previously published by Wen et al., J. Immunological Methods, 2008 which also uses yeast display to identify MHC class II binding peptides derived from the influenza virus genome.

We thank the reviewer for pointing out this reference, as it provides an excellent comparison point for our work. The Wen et al., manuscript utilized an MHC stabilization-based assay similar to other works cited in the Introduction, and analogous to our approach. To our knowledge, the Wen et al., manuscript is the first demonstrated example of yeast display as a means to identify and analyze pathogen-derived peptides presented by class II MHC molecules. They additionally demonstrate that yeast can be used as the sources of antigen for T cell activation assays. There are two important technical advances presented in our work as compared to the Wen at al approach:

– First, to generate libraries, Wen et al., digest amplified viral genomes, which could be advantageous when studying an unsequenced genome of interest, but is a less controlled method of creating a proteome-derived library than pooled oligo synthesis. This is due to efficiency (a digested library will include all three reading frames and both orientations, leading to 5/6 of all sequences in the yeast displayed library not encoding pathogen-derived peptides), as well as ensuring depth of coverage (pooled oligo libraries can ensure essentially quantitative coverage of all possible peptides with single amino acid granularity).

– Second, the Wen at al manuscript relied upon assessment of individual yeast clones for sequence and function. The adoption of next-generation sequencing enables our manuscript to have a comprehensive and granular view of peptide enrichment. This, paired with the enrichment analyses described in our manuscript, enables a much more comprehensive view of the possible class II MHC-binding peptides, even weaker binders.

We have incorporated a reference to the work from Wen et al., and discussion of the technical differences of this approach, in our Introduction and Discussion.

2) A discussion about the significance of the observation that the yeast display-based approach identifies MHC II ligands that are not found by NetMHCIIpan4.0 would enhance the paper. This is an important finding, on the one hand, because the method may provide new training data that will improve computational prediction accuracy. On the other hand, many of these sequences are low-affinity binders and may not be immunoreactive as peptide affinity drives T cell response (e.g., PMID: 16039577, PMID: 31253788). How this fits in the context of the oft-heard criticism that computational approaches overpredict would benefit the discussion, as well.

We have expanded our Discussion to highlight these important discussion points. Specifically, we highlight caveats around low affinity MHC-binding peptides, including effects on immunodominance, as well as examples where low affinity peptides have proven relevant. We also add to our discussion of the utility of these data, emphasizing the potential use of yeast display datasets for augmenting current training data and importance of identifying algorithmic false positives.

3) Related to the observation referenced above in (2), the authors imply in parts (Abstract, Introduction) that the yeast display method is superior to computational predictions because it identifies MHC II ligands not discovered by computational algorithms. However, the current study is limited to three MHC II alleles, examines only one predictor, and does not provide evidence of T cell validation (nor even discussion) of the SARS-CoV-2 and dengue datasets in the context of published predictions, MHC II binding data, and immunological studies. The balanced approach taken in the Discussion where experimental and computational approaches are said to complement each other is constructive as it recognizes that both methods have advantages and disadvantages and is a good model for portraying their relationship in earlier parts of the paper.

We thank the reviewers for this important feedback, and have tempered our language in the Abstract and Introduction to be more balanced, emphasizing how experimental and computational approaches complement one another.

4) There is also one issue of the presentation that requires correction. In Figure 1c, the darker shade of blue in the Round 4 flask in the doped library that represents an increase in enrichment from Round 3 to Round 4 is not matched by the decrease in myc staining in Round 4 relative to Round 3.

We have changed the color of the doped library flasks to reflect the convergence of the of the library and decrease in myc staining, as pointed out.